# CarBoN: Calibrated Best-of-N Sampling Improves Test-time Reasoning

## Abstract

Allocating more computation during inference time (test-time scaling) improves language model performance, especially for reasoning tasks. However, popular methods like Best-of-$N$ sampling often show diminishing returns as $N$ increases. To address this inefficiency, we introduce a general **test-time calibration framework** that adaptively modifies the model toward high-reward reasoning paths, with theoretical guarantees of improving the lower bound of expected reward under finite sampling, all without large language model (LLM) retraining. Within this framework, we propose **CarBoN** (Calibrated Best-of-$N$), a two-phase method that first explores the solution space and then learns a calibration of the logits via an input-specific temperature $T$ and additive shift vector $\delta$, guiding generation toward more reliable reasoning. Experiments on MATH-500 and AIME-2024 show that CarBoN improves efficiency, with up to $4\times$ fewer rollouts to reach the same accuracy, while often achieving higher accuracy under fixed budgets. We also analyze the complementary roles of $T$ and $\delta$ in balancing output diversity and correctness, and demonstrate that the framework also generalizes to step-level sampling strategies such as beam search.

## 1 Introduction

Test-time scaling (TTS) is a practical alternative to ever-larger training, enabling models to "think longer" at inference by allocating additional computation to reasoning. Methods such as chain of thought (OpenAI, 2024; Guo et al., 2025), sequential reasoning (Wang et al., 2022; Qu et al., 2024; Shinn et al., 2023), and parallel sampling (Snell et al., 2024; Beeching et al.; Puri et al., 2025; Liu et al., 2025) demonstrate that increased test-time effort consistently improves performance without retraining. As these studies suggest, TTS allows smaller LLMs to match or even outperform larger ones, providing a more cost-efficient and flexible inference strategy.

Despite these benefits, simply increasing test-time compute does not guarantee optimal performance. Recent work has shown that inference without effective verification is often sub-optimal, as models may spend additional computation on low-quality reasoning paths (Setlur et al., 2025). To overcome this inefficiency, we propose a general **test-time calibration framework** that strategically reallocates the inference budget by leveraging feedback from a verifier or reward model during inference. Rather than treating generation as a fixed forward pass, the model adaptively steers toward high-reward (likely correct) regions, improving reasoning reliability under a fixed query budget.

**Why calibration for TTS? A motivating example of reward-based binary search.**  Let the task be finding a target in $[0, 10^4]$. Calibration means that before each search step the model can query $n$ candidate points for reward, where $n$ denotes the number of reward queries per step. The reward is defined as the inverse distance to the target plus noise. The baseline binary search, corresponding to naive TTS ($n = 0$), requires 13.3 steps on average. Increasing $n$ significantly accelerates convergence: for example, with $n = 16$ the search depth is reduced by up to $74\%$ (see Figure 1, left). Figure 1 (right) shows an example run where calibration quickly converges to the target, while vanilla binary search continues oscillating. This example highlights that reward feedback for calibration reshapes the sampling distribution and motivates its use for TTS.

Building on this principle, our framework reuses sampled completions that are normally discarded in parallel sampling methods (Wang et al., 2022; Brown et al., 2024; Snell et al., 2024) to extract

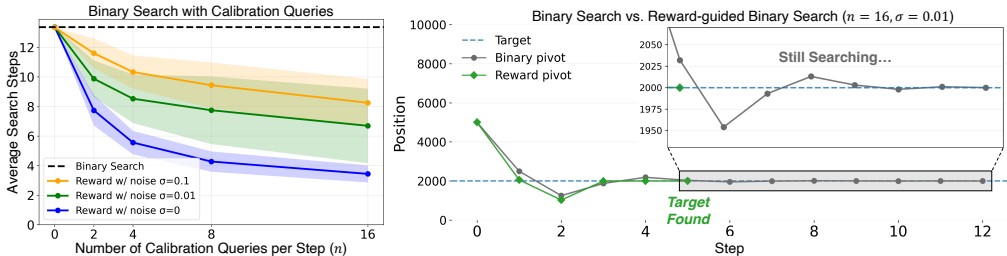

**Figure 1: Reward-guided calibration accelerates binary search.** Left: Increasing per-step noisy reward (inverse-distance signal + noise) lowers average search steps versus vanilla. Right: Example showing reward guidance converges early; vanilla keeps oscillating. See Appendix B.1 for details.

reward signals and perform calibration. Within this framework, we introduce **CarBoN** (Calibrated Best-of-$N$). Without modifying the original LLM, our framework allocates part of the budget to exploration and calibration, then focuses the remaining budget on high-scoring regions using logit calibration. Reusing high-scoring answer selected by reward model enhances answer quality and query efficiency, under the same inference budget.

Our main contributions are summarized as follows:

- **Test-Time Calibration Framework.** We introduce a test-time calibration framework that reallocates the inference budget (Figure 2a). Applied to Best-of-$N$, CarBoN first explores diverse candidates to identify high-scoring regions, then uses logit calibration to focus the remaining budget on high-scoring areas, improving accuracy under fixed rollout budget.

- **Theoretical Guarantees.** We provide formal proofs showing that optimal calibration parameters exist, which improve the expected reward's lower bound under finite sampling and strictly outperform the uncalibrated baseline.

- **CarBoN Empirically Improves Test-Time Reasoning.** Across multiple models and benchmarks, including MATH-500 and the more challenging AIME-2024, CarBoN achieves higher or comparable accuracy with fewer queries than uncalibrated models, showing benefits for both general-purpose and math-specialized models.

- **Calibration Insights and Generalization.** In CarBoN, we find that temperature ($T$) controls output distribution sharpness, delta ($\delta$) corrects token-level biases, and together they balance diversity and correctness to improve test-time reasoning. We further generalize test-time calibration beyond Best-of-$N$, applying to step-level sampling (beam search) to demonstrate broader applicability.

## 2 RELATED WORK

**Reasoning with Intermediate Steps.** Recent work has improved LLM reasoning by encouraging generation of intermediate steps, e.g., chain-of-thought (Wei et al., 2022; Kojima et al., 2022), least-to-most prompting (Zhou et al., 2022), and learned reasoning policies (Yue et al., 2023; Yu et al., 2023; Wang et al., 2023; OpenAI, 2024; Anthropic, 2025; Guo et al., 2025). These methods use a large token budget for   multi-step reasoning within a single forward pass, effectively "thinking longer" at inference. However, they remain limited by context window and KV cache constraints, which can restrict feasible reasoning length and make naive scaling of token generation inefficient.

**Iterative Refinement.** Sequential revision methods improve outputs by feeding previous answers back. Recursive reasoning (Qu et al., 2024) uses multiple critique rounds to correct mistakes; the authors note early errors can propagate and gains often diminish after a few iterations. Reflective prompting (Shinn et al., 2023) adds self-assessment but its effectiveness is limited by memory and reflection quality. Overall, these methods enhance accuracy without retraining but increase latency, RAM usage, and computation linearly, and repeated refinement may yield diminishing returns.

**Parallel Sampling Strategies.** Parallel sampling methods can be divided into two groups. The first generates complete candidate answers per query without intermediate evaluation. This includes self-consistency (majority voting) and best-of-$N$ (BoN), where the former selects the most frequent answer and the latter scores each candidate with a reward model, choosing the highest-scoring answer (Wang et al., 2022; Brown et al., 2024). Best-of-$N$ generally outperforms majority voting,

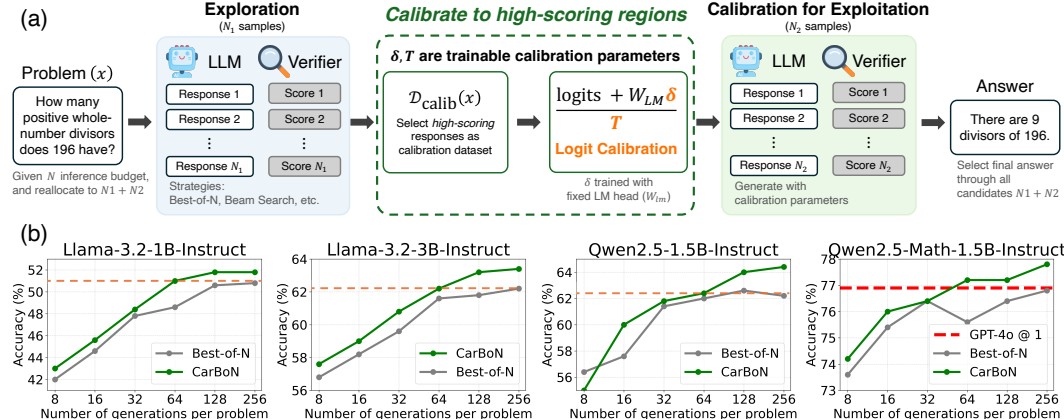

**Figure 2: (a) Test-time calibration framework.** With a rollout budget $N = N_1 + N_2$, the model first explores by generating and scoring $N_1$ candidate responses. The model then learns calibration parameters $(\delta, T)$ from high-scoring responses, , using them to adjust the logits for the remaining $N_2$ generations. The final answer is selected from all $N$ candidates. **(b) MATH-500 Results.** CarBoN improves weighted Best-of-N accuracy across four models. For all models, calibrated accuracy at $N = 64$ (orange dash line) matches or exceeds uncalibrated accuracy at $N = 256$, corresponding to up to a $4\times$ reduction in rollout budgets. Notably, with Qwen2.5-Math-1.5B-Instruct at $N = 64$, CarBoN surpasses GPT-4o (red dashed line), while uncalibrated Best-of-N with $N = 256$ does not.

as verifier-free selection is suboptimal (Setlur et al., 2025). This approach is simple, efficient, and provides a practical baseline.

The second group consists of step-level methods, which evaluate candidates at each generation step for finer-grained control and typically higher-quality results. These include Beam Search (Snell et al., 2024), Diverse Verifier Tree Search (DVTS) (Beeching et al.), and Particle Filtering (Puri et al., 2025), all of which are variations of Beam Search. Beam Search maintains the top-k high-scoring beams at each step. DVTS allows independent beams and optionally samples lower-scoring steps, balancing exploration and exploitation. Particle Filtering converts step scores into probabilities and samples candidate steps, maintaining a diverse particle set for probabilistic inference.

Step-level methods often improve quality and diversity but are computationally costly due to repeated scoring and pruning. In this work, we focus on Best-of-$N$ as the main baseline for test-time calibration and include Beam Search experiments to validate the step-level framework.

**Model Calibration.** Model calibration traditionally aligns a model's predicted probabilities with empirical correctness in classification, using techniques such as temperature scaling (Guo et al., 2017), histogram binning (Zadrozny & Elkan, 2001), isotonic regression (Zadrozny & Elkan, 2002), Dirichlet calibration (Kull et al., 2019), and joint input-output calibration (Tang et al., 2024). These methods generally operate post-hoc on fixed models to improve confidence estimation. In this work, we adopt the idea of post-hoc logits adjustment under a frozen LLM, but change the objective from correctness alignment to calibrating test-time scaling sampling, thereby shifting generation toward higher-reward regions without model retraining or relying on ground-truth labels.

## 3 TEST-TIME CALIBRATION

Building on our motivation in the introduction, we observe that parallel sampling typically generates many candidate completions, of which only the highest-scoring is selected while the rest are discarded. We hypothesize that the discarded completions contain valuable signals that, if reused, can better calibrate the model's output and improve answer quality. This leads to our first research question (RQ): **RQ1.** *How can we guide LLM inference under test-time scaling by reusing information from discarded completions in parallel sampling in order to calibrate the output distribution and enhance answer quality under a fixed compute budget?*

### 3.1 Balancing Exploration and Exploitation at Test-Time

To address RQ1, we first define logits calibration as a learnable transformation that reshapes the model's output distribution at test time. Formally, let $x$ be the input problem, $y = (y_1, \ldots, y_T)$ a generated answer sequence, and $\theta$ the fixed LLM parameters. The calibrated next-token distribution is then defined as:

$$p_\theta(y_t \mid y_{<t}, x; \delta, T) = \text{softmax}\left(\frac{\text{logits} + W_{\text{LM}} \cdot \delta}{T}\right), \tag{1}$$

where $\text{logits} \triangleq f_\theta(x, y_{<t})$ are the base logits for predicting $y_t$ given the input $x$ and prefix $y_{<t}$, $\delta \in \mathbb{R}^d$ is an additive shift vector, and $T > 0$ is a temperature parameter, both learned for calibration at test time. $W_{LM} \in \mathbb{R}^{V \times d}$ denotes the fixed language model head ($\text{lm\_head}$) mapping the last hidden states of dimension $d$ to logits over a vocabulary of size $V$. Since the model is autoregressive, this calibrated distribution adjusts the prediction of the current token and may propagate effects to future token predictions, influencing the generated sequence.

Building on this logits calibration, we design a two-phase optimization framework for test-time inference. Specifically, we split the given inference (rollout) budget $N = N_1 + N_2$. we first use $N_1$ to explore the output space and identify a high-scoring (high-reward) distribution, then calibrate the model's logits to this distribution and use the remaining $N_2$ for focused exploitation (see Figure 2a).

**Phase 1: Exploration ($N_1$ samples).** The model generates $N_1$ candidate answers from the uncalibrated distribution $p_\theta(y \mid x; \delta = 0, T_{\text{base}})$, where $\delta = 0$ and $T_{\text{base}}$ indicate no calibration is applied in this phase. Each candidate is scored by a process reward model (PRM), and these scores are used to identify promising high-reward directions.

**Calibrate to high-scoring regions.** From the exploration results, the top-$k$ highest-scoring completions are selected as the calibration dataset $\mathcal{D}_{\text{calib}}(x) = \{y^{(i)}\}_{i=1}^k$. The calibration parameters $(\delta, T)$ are then optimized on this problem to shift the model's logits toward high-reward regions (see Section 3.2 for training details).

**Phase 2: Calibration for Exploitation ($N_2$ samples).** Using the learned calibration parameters $(\delta^*, T^*)$, the model generates $N_2$ candidates. These samples are focused on the high-reward regions identified during exploration, increasing the likelihood of obtaining correct or high-quality solutions.

It is noteworthy that the final answer is selected from the union of all $N_1 + N_2$ candidates, since the ground truth is unknown during inference. The exploration phase does not guarantee correctness for any individual sample, but it efficiently identifies regions in the output space that are likely to contain high-reward or plausible solutions. The exploitation phase then intensifies sampling within these regions, providing a principled balance between exploration (breadth) and exploitation (focus) under a fixed inference budget.

Let $R(x, y)$ denote the reward score assigned by the process reward model (PRM) to completion $y$ for input $x$. The expected reward under test-time calibration can be decomposed as:

$$\mathbb{E}[R_{\text{final}}] = \mathbb{E}\left[\max_{y \in \mathcal{Y}_{\text{explore}} \cup \mathcal{Y}_{\text{exploit}}} R(x, y)\right], \tag{2}$$

where $\mathcal{Y}_{\text{explore}}$ and $\mathcal{Y}_{\text{exploit}}$ are the $N_1$ exploration and $N_2$ exploitation samples, respectively. This simple formulation highlights the exploration–exploitation tradeoff: exploration samples help cover diverse regions of the output space, while exploitation focuses on high-reward areas, jointly influencing the final maximum reward.

### 3.2 Training $\delta$ and $T$ for Test-time Calibration

To guide the model toward high-reward outputs, we introduce two input-specific test-time calibration parameters: an additive shift vector $\delta \in \mathbb{R}^d$ and a temperature scaling factor $T > 0$. For each input, the learned shift vector $\delta$ is projected through the fixed language model head $W_{LM}$ to produce a token-specific bias in logit space. To reduce dimensionality and avoid overfitting, $\delta$ is trained in a lower-dimensional last hidden space $\mathbb{R}^d$ and mapped via $W_{\text{LM}}$, since directly learning a full logits-space bias would be extremely high-dimensional ($V \gg d$). The temperature $T$ is also learned for each input, providing control over the sharpness of the distribution, where lower $T$ concentrates

probability mass and higher $T$ flattens it. Together, $(\delta, T)$ provide fine-grained alignment toward high-reward completions without modifying the base model parameters.

To efficiently learn the calibration parameters $(\delta, T)$, we leverage the top-$k$ high-reward candidates from the exploration phase, scored by the PRM, as a calibration set $\mathcal{D}_{\text{calib}}(x) = \{y^{(i)}\}_{i=1}^{k}$. Instead of repeatedly performing forward passes through the full model during calibration, we cache the base logits $f(x, y_{<t})$ for each prefix $y_{<t}$ of the high-reward candidates. The calibration parameters $(\delta, T)$ are then optimized directly on these cached logits, making training lightweight and efficient.

$$(\delta^*, T^*) = \arg \min_{\delta, T > 0} \mathbb{E}_{y \sim \mathcal{D}_{\text{calib}}(x)} \left[ -\log p_\theta(y \mid x; \delta, T) \right] + \lambda_\delta \|\delta\|_2^2, \tag{3}$$

where $p_\theta(y \mid x) = \prod_{t=1}^{T} p_\theta(y_t \mid y_{<t}, x)$ factorizes the sequence probability over tokens, and $p_\theta(y \mid x; \delta, T)$ applies the additive shift $\delta$ and temperature $T$ at each token. The regularization coefficient $\lambda_\delta$ mitigates overfitting to the limited calibration set. This calibration captures token-level effects of $(\delta, T)$, enabling input-specific adjustments under a fixed inference budget. Further insights into the roles of $T$ and $\delta$ in calibration are provided in Sec. 6.1.

## 4 THEORETICAL ANALYSIS

Building on the previous section, we now provide theoretical foundations for test-time calibration. We focus on answering the following question: **RQ2.** *Can we provide provable guarantees that test-time calibration improves the expected reward under finite sampling?* To answer this, we first establish the existence of a joint calibration solution and then show that it provably improves the expected reward under Best-of-$N$ sampling.

### 4.1 EXISTENCE OF JOINT CALIBRATION SOLUTIONS

We begin by proving the existence of a joint calibration solution $(\delta^*, T^*)$ that strictly increases the probability of generating a high-quality output, proceeding by construction to show that a non-trivial $\delta$ and $T$ can always beneficially alter the output distribution.

**Lemma 1** (Existence of an Improving Joint Solution $(\delta, T)$)**.** *Let the joint loss function be $\mathcal{L}(\delta, T) = \mathbb{E}_{y \sim \mathcal{D}_{calib}(x)} \left[ -\log p_\theta(y \mid x; \delta, T) \right]$. Let $\bar{p}_\theta$ be the model's average predictive distribution and $\bar{p}_{target}$ be the empirical average one-hot distribution, both averaged over all generation steps in the calibration set $\mathcal{D}_{calib}(x)$. Suppose the base model is not perfectly calibrated in the sense that at least one of the following conditions holds: (1) $\bar{p}_\theta \neq \bar{p}_{target}$, or (2) the average logit of ground-truth tokens does not equal the average expected logit. Then there exists a joint solution $(\delta, T) \in \mathbb{R}^D \times (0, \infty)$, where $(\delta, T) \neq (\mathbf{0}, 1)$, such that the loss is strictly reduced: $\mathcal{L}(\delta, T) < \mathcal{L}(\mathbf{0}, 1)$*

*Proof.* The proof is given in Appendix A.1.

### 4.2 EXPECTED REWARD IMPROVEMENT FROM CALIBRATION

Building on the existence guarantee, we next prove that applying such a joint calibration improves the expected reward under finite Best-of-$N$ sampling. This theorem formally establishes the benefit of our calibration method. The proof demonstrates that the calibrated distribution achieves first-order stochastic dominance over the baseline. Intuitively, this means the calibrated process is more likely to generate high-reward outputs, which in turn guarantees a higher expected maximum reward.

**Theorem 2** (Joint Calibration $(\delta, T)$ Improves Expected Reward from Best-of-$N$ Sampling)**.** *Let $p_\theta(y \mid x; \delta, T)$ be the model's probability distribution over outputs $y \in \mathcal{Y}$, parameterized by a calibration vector $\delta$ and a temperature $T$. Let the base model be configured with parameters $(\mathbf{0}, T_{base})$ for some $T_{base} > 0$. Let $R(x, y)$ be a reward function, and assume there exists a unique output $y^* \in \mathcal{Y}$ with a strictly maximum reward, i.e., $r^* = R(x, y^*) > \max_{y \neq y^*} R(x, y) = r_{other\_max}$. We consider cases where joint calibration with parameters $(\delta^*, T^*)$ improves upon the base model by increasing the probability of the unique optimal output, i.e.,*

$$p_\theta(y^* \mid x; \delta^*, T^*) > p_\theta(y^* \mid x; \mathbf{0}, T_{base}).$$

*Then, for any $n \geq 1$ within the remaining inference budget after calibration, the lower bound on the expected best-of-$N$ reward under the jointly calibrated model is strictly greater than that of the*

*base model. Specifically, let $R_{LB}(p) = r^* - (1-p)^n(r^* - r_{other\_max})$ be a valid lower bound for the expected best-of-$N$ reward, where $p$ is the probability of sampling $y^*$. The improvement in this lower bound, $\Delta_{R_{LB}}(x, n) = R_{LB}(p_\theta(y^* \mid x; \delta^*, T^*)) - R_{LB}(p_\theta(y^* \mid x; \mathbf{0}, T_{base}))$, is strictly positive.*

*Proof.* The proof is given in Appendix A.2.

As a direct consequence of this theorem, we present a corollary that provides theoretical justification for our two-phase sampling strategy. During test-time inference, one might be tempted to discard the initial exploration samples and rely only on the "exploited" ones. We show that this strategy is suboptimal, and to maximize the expected reward the final answer should be selected from the combined set of all $N_1$ exploration and $N_2$ exploitation candidates. This result highlights that the exploration phase is essential, contributing irreplaceable value by ensuring the final candidate pool is both broad and targeted.

**Corollary 3** (Sub-optimality of Exploitation Alone). *The final candidate is selected by maximizing $R(x, y)$ over a set of candidates $\mathcal{Y}$. Since $\mathcal{Y}_{exploit}$ is a subset of the union $\mathcal{Y} = \mathcal{Y}_{explore} \cup \mathcal{Y}_{exploit}$, the strategy of only selecting from $\mathcal{Y}_{exploit}$ is sub-optimal compared to selecting from the union. This is because the maximum reward achievable from the union is greater than or equal to the maximum reward achievable from the exploitation set alone.*

$$\max_{y \in \mathcal{Y}_{explore} \cup \mathcal{Y}_{exploit}} R(x, y) \geq \max_{y \in \mathcal{Y}_{exploit}} R(x, y)$$

*Proof.* The proof is given in Appendix A.3.

In summary, these results affirmatively answer RQ2, showing that effective test-time calibration is achievable and beneficial.

## 5 RESULTS

### 5.1 EXPERIMENTAL SETUP

**Models.** We evaluate Llama-3.2-1B/3B-Instruct (Meta AI, 2024) and Qwen2.5-1.5B-Instruct / Qwen2.5-Math-1.5B/7B-Instruct (Qwen Team, 2024; Yang et al., 2024), all in bf16. These include general-purpose (Llama, Qwen2.5) and math-specialized (Qwen2.5-Math) models, providing diverse capabilities for calibration evaluation.

**Process Reward Model (PRM).** All experiments use Qwen2.5-Math-PRM-7B (Zhang et al., 2025), a state-of-the-art reward model for mathematical reasoning. It assigns step-level scores (0–1) to intermediate reasoning steps, enabling fine-grained evaluation beyond final answers. Following Zhang et al. (2025), we adopt the reward of the final step (*last score*) as the overall score, which outperforms product and minimum strategies for PRMs trained via Monte Carlo estimation.

**Baseline Setup.** We set $T = 0.8$ for the baseline best-of-$N$ method, as this value achieves the overall best results in a grid search over $[0.1, 1.6]$ and is consistent with previous studies (Snell et al., 2024). For a comprehensive analysis of temperature effects, see Appendix C.2.

**Calibration Training.** Calibration parameters $(\delta, T)$ are optimized on cached logits using the top-$k$ high-scoring completions from an initial $N_1 = N/2$ runs ($T = 0.8$) as the calibration dataset. The remaining $N_2 = N/2$ completions are generated using the learned parameters. This two-stage procedure is lightweight, requires no additional inference, and learns $\delta$ and $T$ at test time for each input. More details are provided in Appendix C.3.

**Dataset.** We use the MATH benchmark (Hendrycks et al., 2021), covering high-school level competition problems of varying topics and difficulty. Experiments are conducted on the MATH-500 test split (Lightman et al., 2023), widely adopted for evaluating LLM mathematical reasoning. Additionally, we include AIME-2024 (HuggingFaceH4, 2024), a smaller and more challenging dataset (30 problems/year), evaluated using the math-specialized Qwen2.5-Math models (1.5B and 7B).

**Evaluation Metric.** Accuracy is the proportion of completions whose final answers exactly match the ground truth. For *vanilla*, the highest-scoring completion among $N$ candidates is selected. For *weighted*, PRM scores for identical answers are summed and the answer with the highest aggregated score is chosen. All comparisons are made with the same rollout (inference) budget $N$.

**Table 1: Accuracy (%) of four models on MATH-500, comparing Weighted Best-of-$N$ methods before and after calibration.** CarBoN enables further improvements beyond the plateau of standard Best-of-$N$, with calibrated accuracy at $N = 64$ exceeding the uncalibrated results at $N = 256$, corresponding to up to $4\times$ less rollout budgets. Bold indicates better accuracy for each $N$.

| Model | Method | N | | | | | |
|---|---|---|---|---|---|---|---|
| | | 8 | 16 | 32 | 64 | 128 | 256 |
| Llama-3.2-1B-Instruct | Best-of-$N$ | 42.0 | 44.6 | 47.8 | 48.6 | 50.6 | 50.8 |
| | CarBoN | **43.0** | **45.6** | **48.4** | **51.0** | **51.8** | **51.8** |
| Llama-3.2-3B-Instruct | Best-of-$N$ | 56.8 | 58.2 | 59.6 | 61.6 | 61.8 | 62.2 |
| | CarBoN | **57.6** | **59.0** | **60.8** | **62.2** | **63.2** | **63.4** |
| Qwen2.5-1.5B-Instruct | Best-of-$N$ | **56.4** | 57.6 | 61.4 | 62.0 | 62.6 | 62.2 |
| | CarBoN | 55.0 | **60.0** | **61.8** | **62.4** | **64.0** | **64.4** |
| Qwen2.5-Math-1.5B-Instruct | Best-of-$N$ | 73.6 | 75.4 | **76.4** | 75.6 | 76.4 | 76.8 |
| | CarBoN | **74.2** | **76.0** | **76.4** | **77.2** | **77.2** | **77.8** |

## 5.2 CARBON: CALIBRATED BEST-OF-$N$ IMPROVES ACCURACY AND EFFICIENCY

We evaluate CarBoN, which applies test-time calibration to the Best-of-$N$ strategy, on different LLMs using the MATH-500 benchmark. We report Weighted results in Table 1 (full tables including Vanilla are in Appendix B.2), showing similar trends with greater stability across models and $N$.

For large rollout $N$ (64, 128, 256), uncalibrated Best-of-$N$ results plateau, yielding minimal gains. Llama-3.2-3B-Instruct improves only 0.6% from $N = 64$ to 256, and Qwen2.5-1.5B-Instruct gains between 0.2% and 0.6%, peaking at $N = 128$. In contrast, CarBoN continues to improve performance beyond this limit. For example, all models achieve higher accuracy at $N = 64$ with CarBoN than the uncalibrated baseline at $N = 256$, reducing the required rollout budgets by up to $4\times$. Notably, Qwen2.5-Math-1.5B-Instruct with CarBoN at $N = 64$ reaches 77.2% accuracy, surpassing GPT-4o @1 (77.0%; see Appendix B.3 for more details and results on larger models), while the uncalibrated Best-of-$N$ at $N = 256$ reaches only 76.8%.

At $N = 256$, CarBoN improves Weighted decoding by over 1% across all models. Vanilla decoding also shows notable gains, up to 3.6% for Qwen2.5-1.5B-Instruct. These results show that CarBoN not only increases accuracy but also reduces sampling costs.

We also experiment with larger math-specialized models (Qwen2.5-Math-7B-Instruct) and a more challenging benchmark, AIME-2024 (HuggingFaceH4, 2024), which contains 30 high-difficulty problems. Table 2 reports the number of correct answers for both Qwen2.5-Math-1.5B/7B-Instruct across different rollout budgets $N$. Even on this small and difficult dataset, CarBoN improves over the uncalibrated Best-of-$N$, demonstrating that test-time calibration boosts performance for larger models and harder problems while requiring fewer rollouts.

Overall, these results show that CarBoN, which is a concrete instance of the test-time calibration framework, consistently improves reasoning performance and enhances the quality of selected outputs without modifying the underlying decoding strategy.

**Table 2: Correct answers (out of 30) on the AIME-2024 benchmark for two math-specialized models, comparing Best-of-$N$ and CarBoN across different rollout budgets.** CarBoN enables further improvements beyond the plateau of standard Best-of-$N$. Bold numbers indicate the higher number of correct answers for each $N$.

| Model | Method | N | | | | |
|---|---|---|---|---|---|---|
| | | 16 | 32 | 64 | 128 | 256 |
| Qwen2.5-Math-1.5B-Instruct | Best-of-$N$ | 4/30 | 5/30 | 6/30 | 6/30 | 6/30 |
| | CarBoN | 4/30 | 5/30 | 6/30 | **7/30** | **7/30** |
| Qwen2.5-Math-7B-Instruct | Best-of-$N$ | 5/30 | 5/30 | 6/30 | 6/30 | 6/30 |
| | CarBoN | 5/30 | **6/30** | 6/30 | 6/30 | **7/30** |

## 5.3 Effect of the Calibration Parameters $(\delta, T)$

We perform an ablation study on a general-purpose model (Llama-3.2-1B-Instruct) and a math-specialist model (Qwen2.5-Math-1.5B-Instruct) to isolate the contributions of the additive shift $\delta$ and temperature $T$ (full tables including Vanilla decoding are in Appendix B.2). For each experiment, we retrain calibration with only one parameter enabled, isolating the effect of $\delta$ or $T$.

Table 3 shows that adding $\delta$ alone already improves over the baseline once $N$ is sufficiently large, while the combination of $\delta$ and $T$ (CarBoN) yields the strongest gains. For instance, CarBoN reaches 51.8% for Llama-3.2-1B-Instruct and 77.8% for Qwen2.5-Math-1.5B-Instruct. These results indicate that using $\delta$ and $T$ together provides the most reliable gains. In the next section, we analyze how $\delta$ and $T$ contribute to answer quality.

## 6 Discussion of Calibration and Generalization

Beyond the main results and ablation studies, we further analyze the distinct roles of temperature $T$ and $\delta$, and examine the generalization of test-time calibration beyond Best-of-$N$ sampling.

### 6.1 How Token-level Calibration Improves Answer Quality

**Temperature Adaptation.** We find that calibration temperature strongly correlates with problem difficulty. In Figure 3 (with $N_1 = 128$ for exploration, $k = 32$ for calibration), harder questions have higher temperatures, since greater diversity improves the chance of reaching correct answers. To explain this, we analyze the entropy of the top-$k$ high-scoring completions used for calibration. The blue curve shows entropy rising with difficulty, meaning the model produces more diverse outputs when less confident. Although calibration only uses high-scoring responses, this diversity remains, enabling the model to learn an appropriate temperature for different difficulty levels.

Beyond difficulty, we also find that the calibration temperature increases with $N$, as higher temperatures promote diversity and better utilize the inference budget (see Appendix B.4). Larger $N$ enables more exploration, while a higher temperature prevents near-identical samples, ensuring that the additional budget contributes meaningfully. This highlights that temperature should adapt to problem difficulty and inference budget, and our calibration achieves this adaptation.

**Delta Adjustment.** We report four overlap metrics with respect to top-$k$ high-scoring answers: Jaccard and Dice (set-level similarity), and Recall and Precision (token-level coverage and specificity), with full definitions in the Appendix B.6. After applying $\delta$ calibration, the generated tokens show higher set-level similarity, as measured by Jaccard and Dice, and increased token-level Precision, with the results for each metric comparing the calibrated and uncal-

**Table 4:** Token-level overlap with/without $\delta$ calibration against top-$k$ high-scoring answers on Llama-3.2-1B-Instruct ($N_1 = 128, k = 32$).

| Metric | Calibration w/ $\delta$ | No Calibration |
|---|---|---|
| Jaccard Overlap ↑ | **0.5184** | 0.4838 |
| Dice Overlap ↑ | **0.6805** | 0.6497 |
| Token Recall ↑ | 0.8804 | **0.8919** |
| Token Precision ↑ | **0.5590** | 0.5147 |

**Table 3: Ablation study on calibration parameters** $(\delta, T)$ **and their combination (CarBoN) for Best-of-**$N$ **search on MATH-500.** We compare applying a shift ($\delta$), a temperature scaling ($T$), and their joint calibration (CarBoN) under Weighted selection. All values report accuracy (%). Results show that CarBoN consistently improves accuracy across different $N$, highlighting the complementary benefits of $\delta$ and $T$. Bold numbers indicate the better accuracy for each $N$.

| Model | Method | N | | | | | |
|---|---|---|---|---|---|---|---|
| | | 8 | 16 | 32 | 64 | 128 | 256 |
| Llama-3.2-1B-Instruct | Best-of-$N$ | 42.0 | 44.6 | 47.8 | 48.6 | 50.6 | 50.8 |
| | Best-of-$N$ w/ $\delta$ | 41.8 | 43.8 | 48.2 | 49.0 | 51.0 | 51.2 |
| | Best-of-$N$ w/ $T$ | 42.0 | 45.0 | 47.0 | 49.6 | 49.8 | 50.6 |
| | CarBoN | **43.0** | **45.6** | **48.4** | **51.0** | **51.8** | **51.8** |
| Qwen2.5-Math-1.5B-Instruct | Best-of-$N$ | 73.6 | 75.4 | **76.4** | 75.6 | 76.4 | 76.8 |
| | Best-of-$N$ w/ $\delta$ | **74.2** | 74.8 | 75.6 | 77.0 | 76.6 | 77.0 |
| | Best-of-$N$ w/ $T$ | 73.2 | 75.2 | 76.0 | 76.4 | 76.0 | 76.6 |
| | CarBoN | **74.2** | **76.0** | **76.4** | **77.2** | **77.2** | **77.8** |

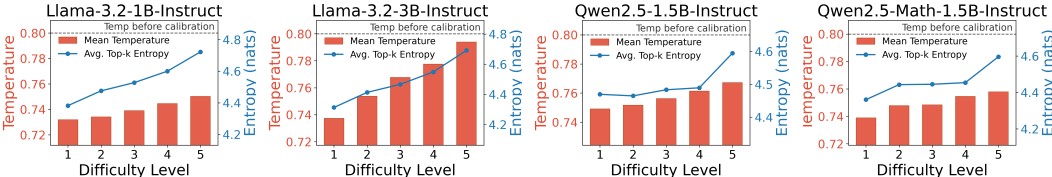

**Figure 3: Correlation between problem difficulty, calibrated temperature, and top-$k$ completion entropy on MATH-500.** Bars (left y-axis) show the average learned temperature across five difficulty levels, while the line plot (right y-axis) shows the normalized entropy of the top-$k$ completions used for calibration. Both temperature and entropy strongly increase with problem difficulty, indicating that harder problems require higher temperatures to capture the more diverse top-$k$ token distributions. See Appendix B.5 for full Spearman correlation statistics.

ibrated settings presented in Table 4, indicating that the model's generations are more aligned with the patterns of high-quality responses. These results are consistent with our ablation study in Table 3, where incorporating $\delta$ improves task-level accuracy, particularly when aggregating a larger candidate pool, suggesting that $\delta$ effectively guides the model toward high-scoring behavior.

## 6.2 GENERALIZING TEST-TIME CALIBRATION BEYOND BEST-OF-$N$

While test-time calibration improves reasoning ability in Best-of-$N$ and offers greater efficiency, other step-level strategies score each reasoning step before proceeding to the next. While such methods (e.g., beam search, DVTS, and particle filtering) can achieve higher performance when more fine-grained guidance is available during generation, they need heavier computation due to repeated verifier calls. To illustrate that test-time calibration can generalize beyond Best-of-$N$, we focus on beam search as a representative step-level sampling strategy.

As shown in Table 5, calibrated beam search provides improvements over the standard baseline in most settings across both models. Notably, with $N = 32$, the calibrated beam search reaches accuracy close to or even matching that of beam search with $N = 64$, indicating that calibration improves sample efficiency by reducing the number of candidates required to achieve a given performance level. This demonstrates that test-time calibration is not limited to Best-of-$N$ but can also enhance fine-grained step-level decoding, suggesting a promising direction for integrating calibration with step-level sampling methods.

**Table 5: Accuracy (%) of standard and calibrated beam search on the MATH-500 benchmark.** Calibrated beam search generally improves test-time reasoning performance, especially for larger $N$.

| Model | Method | N | | | |
|---|---|---|---|---|---|
| | | 8 | 16 | 32 | 64 |
| Llama-3.2-1B-Instruct | Beam Search | 56.0 | 58.4 | 60.4 | 62.2 |
| | Calibrated Beam Search | **57.2** | **60.0** | **62.2** | **64.2** |
| Qwen2.5-Math-1.5B-Instruct | Beam Search | **79.0** | 79.2 | 80.2 | 81.4 |
| | Calibrated Beam Search | 78.6 | **79.6** | **81.2** | **82.8** |

## 7 CONCLUSION

We introduced test-time calibration, a framework to adapt LLMs at inference under test-time scaling via additive logits shifts and adaptive temperature scaling, instantiated as CarBoN on Best-of-N. Our theoretical analysis shows calibration can provably improve accuracy and the lower bound of expected reward under finite samples. Empirically, CarBoN consistently improves performance across benchmarks, rollout budgets, and step-wise sampling, demonstrating its generalization potential. We believe this framework will inspire and advance future designs of test-time scaling methods.

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

## LLM USAGE DISCLOSURE

We used Large Language Models to assist in writing and coding for this paper. ChatGPT and Gemini were employed to help polish language, improve clarity, and refine expression. GitHub Copilot was used to provide autocomplete suggestions and minor code snippets. All core ideas, designs, and conclusions were independently developed and verified by the authors.

## REPRODUCIBILITY STATEMENT

To ensure reproducibility, we provide detailed descriptions of our methodology, theoretical derivations, and experimental setup in the paper and appendix. The code used for experiments is included in the supplementary materials to support verification and replication of our results.

## A  MATHEMATICAL PROOFS

### A.1  PROOF OF LEMMA 1

**Lemma 1** (Existence of an Improving Joint Solution $(\delta, T)$)  *Let the joint loss function be* $\mathcal{L}(\delta, T) = \mathbb{E}_{y \sim \mathcal{D}_{calib}(x)} [-\log p_\theta(y \mid x; \delta, T)]$. *Let* $\bar{p}_\theta$ *be the model's average predictive distribution and* $\bar{p}_{target}$ *be the empirical average one-hot distribution, both averaged over all generation steps in the calibration set* $\mathcal{D}_{calib}(x)$. *Suppose the base model is not perfectly calibrated in the sense that at least one of the following conditions holds: (1)* $\bar{p}_\theta \neq \bar{p}_{target}$, *or (2) the average logit of ground-truth tokens does not equal the average expected logit. Then there exists a joint solution* $(\delta, T) \in \mathbb{R}^D \times (0, \infty)$, *where* $(\delta, T) \neq (\mathbf{0}, 1)$, *such that the loss is strictly reduced:*

$$\mathcal{L}(\delta, T) < \mathcal{L}(\mathbf{0}, 1)$$

*Proof.* The proof demonstrates that the joint loss function $\mathcal{L}(\delta, T)$ is continuously differentiable and that its gradient, evaluated at the initial point $(\delta, T) = (\mathbf{0}, 1)$, is a non-zero vector. For a continuously differentiable function, a non-zero gradient at a point guarantees the existence of a strict descent direction, ensuring that a nearby point with a lower loss value exists.

**Continuity and Differentiability.**  The loss $\mathcal{L}(\delta, T)$ is the average of the per-step negative log-likelihoods (NLL) $L_{i,j}(\delta, T)$ over the calibration set. The per-step NLL is a function of the logits, which are an affine transformation of $\delta$, divided by temperature $T$, and then passed through a log-softmax function. Specifically, $L_{i,j}(\delta, T) = -\log(\text{softmax}((W_{\text{lm}}(h_{i,j} + \delta))/T))_{y_{i,j}}$. Since affine transformations, division by a non-zero scalar, the exponential function, and the logarithm function are all continuously differentiable ($C^\infty$) on their domains, their composition, the log-softmax, is also continuously differentiable. As $\mathcal{L}$ is a finite sum of such functions, it is also continuously differentiable on its domain $\mathbb{R}^D \times (0, \infty)$.

**The Joint Loss Function.**  The loss $\mathcal{L}(\delta, T)$ is the average negative log-likelihood (NLL) over the calibration set $\mathcal{D}_{\text{calib}}(x) = \{y_i\}_{i=1}^K$. Let $n_i$ denote the length of the sequence $y_i$. The NLL for a single answer sequence $y_i = (y_{i,1}, \ldots, y_{i,n_i})$ is the sum of the NLLs for each of its $n_i$ generation steps:

$$-\log p_\theta(y_i \mid x; \delta, T) = \sum_{j=1}^{n_i} -\log p_\theta(y_{i,j} \mid x, y_{i,<j}; \delta, T)$$

The total loss $\mathcal{L}$ is the average of this quantity over all $K$ sequences. Let $N = \sum_{i=1}^K n_i$ be the total number of generation steps across the entire calibration set. The loss can be written as an average over all these steps:

$$\mathcal{L}(\delta, T) = \frac{1}{N} \sum_{i=1}^K \sum_{j=1}^{n_i} L_{i,j}(\delta, T)$$

where $L_{i,j}$ is the NLL for predicting token $y_{i,j}$ given the context $(x, y_{i,<j})$. We then evaluate the gradient $\nabla \mathcal{L}(\delta, T) = \left[ \nabla_\delta \mathcal{L}, \frac{\partial \mathcal{L}}{\partial T} \right]$ at the initial point $(\mathbf{0}, 1)$.

**Evaluating the Gradient Components at** $(\delta, T) = (\mathbf{0}, 1)$**.**

GRADIENT WITH RESPECT TO $\delta$.  At each generation step $(i, j)$, the shift $\delta$ is added to the hidden state $h_{i,j}$ before the final projection: $W_{\text{lm}}(h_{i,j} + \delta)$. The total gradient $\nabla_\delta \mathcal{L}$ is the average of the step-wise gradients. At $(\mathbf{0}, 1)$, this is:

$$\nabla_\delta \mathcal{L}(\mathbf{0}, 1) = \frac{1}{N} \sum_{i=1}^{K} \sum_{j=1}^{n_i} W_{\text{lm}}^\top \left( p_{i,j} - \mathbf{e}_{y_{i,j}} \right) = W_{\text{lm}}^\top \left( \bar{p}_\theta - \bar{p}_{\text{target}} \right) \tag{4}$$

where $p_{i,j} = p_\theta(\cdot \mid x, y_{i,<j})$ is the base model's probability distribution for that step, $\mathbf{e}_{y_{i,j}}$ is the one-hot vector for the target token, $\bar{p}_\theta = \frac{1}{N} \sum_{i,j} p_{i,j}$ is the average predicted distribution, and $\bar{p}_{\text{target}} = \frac{1}{N} \sum_{i,j} \mathbf{e}_{y_{i,j}}$ is the average target distribution.

GRADIENT WITH RESPECT TO $T$.  Let $g_{i,j}$ be the vector of base model logits at step $(i, j)$. The step-wise loss is $L_{i,j}(T) = \log \left( \sum_{k=1}^{C} e^{g_{i,j,k}/T} \right) - \frac{g_{i,j,y_{i,j}}}{T}$. We now derive its partial derivative with respect to $T$:

$$\frac{\partial L_{i,j}(T)}{\partial T} = \frac{\partial}{\partial T} \left[ \log \left( \sum_k e^{g_{i,j,k}/T} \right) \right] - \frac{\partial}{\partial T} \left[ \frac{g_{i,j,y_{i,j}}}{T} \right] \tag{5}$$

$$= \frac{1}{\sum_k e^{g_{i,j,k}/T}} \cdot \left( \sum_k e^{g_{i,j,k}/T} \cdot \frac{-g_{i,j,k}}{T^2} \right) + \frac{g_{i,j,y_{i,j}}}{T^2} \tag{6}$$

$$= \frac{g_{i,j,y_{i,j}}}{T^2} - \frac{1}{T^2} \sum_k \frac{e^{g_{i,j,k}/T}}{\sum_l e^{g_{i,j,l}/T}} \cdot g_{i,j,k} \tag{7}$$

$$= \frac{1}{T^2} \left( g_{i,j,y_{i,j}} - \mathbb{E}_{p_{i,j}(T)}[g_{i,j}] \right) \tag{8}$$

where $\mathbb{E}_{p_{i,j}(T)}[g_{i,j}]$ is the expected logit value under the softmax distribution with temperature $T$.

Evaluating at $T = 1$ and averaging over all steps gives the total gradient for $T$:

$$\left. \frac{\partial \mathcal{L}}{\partial T} \right|_{(\mathbf{0},1)} = \frac{1}{N} \sum_{i=1}^{K} \sum_{j=1}^{n_i} \left( g_{i,j,y_{i,j}} - \mathbb{E}_{p_{i,j}}[g_{i,j}] \right) \tag{9}$$

**The Joint Gradient is Non-Zero.**  The joint gradient is zero only if both of its components are zero.

1. $\delta$-**gradient**: By our premise, $\bar{p}_\theta \neq \bar{p}_{\text{target}}$. The gradient $\nabla_\delta \mathcal{L}(\mathbf{0}, 1)$ is zero only if the non-zero vector $(\bar{p}_\theta - \bar{p}_{\text{target}})$ lies in the null space of $W_{\text{lm}}^\top$. This is equivalent to the error vector being orthogonal to the column space of $W_{\text{lm}}$. For large language models where $D \ll C$, this column space (of dimension at most $D$) is a very small subspace of $\mathbb{R}^C$. It is therefore highly improbable for a specific error vector, arising from model-data mismatch, to lie in the orthogonal complement of this subspace. Thus, we can assert with high confidence that $\nabla_\delta \mathcal{L}(\mathbf{0}, 1) \neq \mathbf{0}$.

2. $T$-**gradient**: The $T$-gradient is zero only if $\frac{1}{N} \sum_{i,j} g_{i,j,y_{i,j}} = \frac{1}{N} \sum_{i,j} \mathbb{E}_{p_{i,j}}[g_{i,j}]$. This condition implies a perfect balance where the average logit of the ground-truth tokens equals the average expected logit over the vocabulary. An uncalibrated model typically exhibits systematic over-confidence (target logit is higher than the average, making the gradient negative) or under-confidence (target logit is lower, making the gradient positive). It is therefore highly unlikely for this gradient to be exactly zero unless the model is already well-calibrated in this specific sense.

Given that the model is not perfectly calibrated, it is guaranteed that at least one of the gradient components is non-zero. Therefore, the joint gradient $\nabla \mathcal{L}(\mathbf{0}, 1)$ is non-zero.

**Existence of an Improving Solution.** A non-zero gradient at the point $(\mathbf{0}, 1)$ implies the existence of a strict descent direction, $-\nabla \mathcal{L}(\mathbf{0}, 1)$. By Taylor's theorem for multivariate functions, for a small step $\alpha > 0$ in this direction, the new point $(\delta', T') = (\mathbf{0}, 1) - \alpha \nabla \mathcal{L}(\mathbf{0}, 1)$ will satisfy $\mathcal{L}(\delta', T') < \mathcal{L}(\mathbf{0}, 1)$. This proves the existence of an improving joint solution.

$\square$

### A.2 PROOF OF THEOREM 2

**Theorem 2** (Joint Calibration $(\delta, T)$ Improves Expected Reward from Best-of-N Sampling) *Let $p_\theta(y \mid x; \delta, T)$ be the model's probability distribution over outputs $y \in \mathcal{Y}$, parameterized by a calibration vector $\delta$ and a temperature $T$. Let the base model be configured with parameters $(\mathbf{0}, T_{base})$ for some $T_{base} > 0$. Let $R(x, y)$ be a reward function, and assume there exists a unique output $y^* \in \mathcal{Y}$ with a strictly maximum reward, i.e., $r^* = R(x, y^*) > \max_{y \neq y^*} R(x, y) = r_{other\_max}$. We consider cases where joint calibration with parameters $(\delta^*, T^*)$ improves upon the base model by increasing the probability of the unique optimal output, i.e.,*

$$p_\theta(y^* \mid x; \delta^*, T^*) > p_\theta(y^* \mid x; \mathbf{0}, T_{base})$$

*Then, for any $n \geq 1$ within the remaining inference budget after calibration, the lower bound on the expected best-of-N reward under the jointly calibrated model is strictly greater than that of the base model. Specifically, let $R_{LB}(p) = r^* - (1-p)^n (r^* - r_{other\_max})$ be a valid lower bound for the expected best-of-N reward, where $p$ is the probability of sampling $y^*$. The improvement in this lower bound, $\Delta_{R_{LB}}(x, n) = R_{LB}(p_\theta(y^* \mid x; \delta^*, T^*)) - R_{LB}(p_\theta(y^* \mid x; \mathbf{0}, T_{base}))$, is strictly positive.*

*Proof.* The proof proceeds in three steps. First, we establish the expression for the expected best-of-N reward and derive its lower bound $R_{LB}(p)$. Second, we prove that this lower bound $R_{LB}(p)$ is a strictly increasing function of $p$, the probability of sampling the optimal output $y^*$. Finally, we use this monotonicity to prove the theorem's main claim.

**Derivation of the Lower Bound** $R_{LB}(p)$**.** Let $p = p_\theta(y^* \mid x)$ be the probability of sampling the unique optimal output $y^*$ in a single trial. The expected best-of-N reward, $\mathbb{E}[\max_{i=1,\ldots,n} R(x, y_i)]$, can be formulated by conditioning on whether $y^*$ is sampled at least once in $n$ trials.

The probability of sampling $y^*$ at least once is $1 - (1-p)^n$. In this event, the maximum reward obtained is exactly $r^*$. The probability of never sampling $y^*$ in $n$ trials is $(1-p)^n$. In this event, the expected maximum reward is $\mathbb{E}_{\text{other}} = \mathbb{E}[\max_{i=1,\ldots,n} R(x, y_i) \mid \forall i, y_i \neq y^*]$.

The total expected reward is thus:

$$\begin{aligned}
\mathbb{E}[\max_{i=1,\ldots,n} R(x, y_i)] &= [1 - (1-p)^n] r^* + (1-p)^n \mathbb{E}_{\text{other}} \\
&= r^* - (1-p)^n (r^* - \mathbb{E}_{\text{other}})
\end{aligned} \tag{10}$$

By definition, $\mathbb{E}_{\text{other}}$ is the expected maximum reward from a set of outputs where none is $y^*$. Therefore, this value cannot exceed the maximum possible reward in that set, $r_{\text{other\_max}}$. This gives the inequality $\mathbb{E}_{\text{other}} \leq r_{\text{other\_max}}$. Since $(r^* - x)$ is a decreasing function of $x$, this implies $(r^* - \mathbb{E}_{\text{other}}) \geq (r^* - r_{\text{other\_max}})$.

Substituting this into the expression for the expected reward yields a valid lower bound, which we denote $R_{LB}(p)$:

$$\mathbb{E}[\max_{i=1,\ldots,n} R(x, y_i)] \geq r^* - (1-p)^n (r^* - r_{\text{other\_max}}) := R_{LB}(p) \tag{11}$$

**Prove the Monotonicity of** $R_{LB}(p)$**.** We now show that $R_{LB}(p)$ is a strictly increasing function of $p$ for $p \in [0, 1)$. We take the derivative of $R_{LB}(p)$ with respect to $p$:

$$\begin{aligned}
\frac{dR_{LB}}{dp} &= \frac{d}{dp}[r^* - (1-p)^n (r^* - r_{\text{other\_max}})] \\
&= -(-n(1-p)^{n-1})(r^* - r_{\text{other\_max}}) \\
&= n(1-p)^{n-1}(r^* - r_{\text{other\_max}})
\end{aligned} \tag{12}$$

By the theorem's assumptions, $n \geq 1$ and $r^* > r_{\text{other\_max}}$, which means $(r^* - r_{\text{other\_max}}) > 0$. For $p \in [0, 1)$, the term $(1 - p)^{n-1}$ is also strictly positive. Therefore, $\frac{dR_{LB}}{dp} > 0$ for all $p \in [0, 1)$, which proves that $R_{LB}(p)$ is a strictly increasing function of $p$.

**Conclusion.** Let $p_{\text{cal}} = p_\theta(y^* \mid x; \delta^*, T^*)$ and $p_{\text{base}} = p_\theta(y^* \mid x; \mathbf{0}, T_{base})$. The theorem's central premise is $p_{\text{cal}} > p_{\text{base}}$. Since $R_{LB}(p)$ is a strictly increasing function of $p$, the inequality $p_{\text{cal}} > p_{\text{base}}$ directly implies:

$$R_{LB}(p_{\text{cal}}) > R_{LB}(p_{\text{base}}) \tag{13}$$

This proves that the lower bound on the expected reward is strictly greater for the calibrated model. The magnitude of this improvement, $\Delta_{R_{LB}}(x, n)$, is given by:

$$\begin{aligned}
\Delta_{R_{LB}}(x, n) &= R_{LB}(p_{\text{cal}}) - R_{LB}(p_{\text{base}}) \\
&= [r^* - (1 - p_{\text{cal}})^n (r^* - r_{\text{other\_max}})] \\
&\quad - [r^* - (1 - p_{\text{base}})^n (r^* - r_{\text{other\_max}})] \\
&= (r^* - r_{\text{other\_max}}) [(1 - p_{\text{base}})^n - (1 - p_{\text{cal}})^n]
\end{aligned} \tag{14}$$

Since $p_{\text{cal}} > p_{\text{base}}$, it follows that $(1 - p_{\text{cal}}) < (1 - p_{\text{base}})$. For $n \geq 1$, this implies $(1 - p_{\text{cal}})^n < (1 - p_{\text{base}})^n$. Thus, the term in the square brackets is strictly positive, and consequently $\Delta_{R_{LB}}(x, n) > 0$.

This completes the proof.

$\square$

### A.3 PROOF OF CORALLARY 3

**Corollary 3** (Sub-optimality of Exploitation Alone) *The final candidate is selected by maximizing $R(x, y)$ over a set of candidates $\mathcal{Y}$. Since $\mathcal{Y}_{exploit}$ is a subset of the union $\mathcal{Y} = \mathcal{Y}_{explore} \cup \mathcal{Y}_{exploit}$, the strategy of only selecting from $\mathcal{Y}_{exploit}$ is sub-optimal compared to selecting from the union. This is because the maximum reward achievable from the union is greater than or equal to the maximum reward achievable from the exploitation set alone.*

$$\max_{y \in \mathcal{Y}_{explore} \cup \mathcal{Y}_{exploit}} R(x, y) \geq \max_{y \in \mathcal{Y}_{exploit}} R(x, y)$$

*Proof.* Let $R^*_{\text{explore}} = \max_{y \in \mathcal{Y}_{\text{explore}}} R(x, y)$ and $R^*_{\text{exploit}} = \max_{y \in \mathcal{Y}_{\text{exploit}}} R(x, y)$. The reward selected from the exploitation set is $R^*_{\text{exploit}}$, while the reward from the combined set is $R_{\text{final}} = \max_{y \in \mathcal{Y}_{\text{explore}} \cup \mathcal{Y}_{\text{exploit}}} R(x, y)$.

By the definition of the maximum function, the maximum of a set is greater than or equal to any of its elements. It follows that for any sampling outcome:

$$R_{\text{final}} = \max(R^*_{\text{explore}}, R^*_{\text{exploit}}) \geq R^*_{\text{exploit}} \tag{15}$$

This inequality holds universally for every possible generated set. By the monotonicity of expectation, taking the expectation over all outcomes yields:

$$\mathbb{E}[R_{\text{final}}] \geq \mathbb{E}[R^*_{\text{exploit}}] \tag{16}$$

This demonstrates that retaining all $N_1 + N_2$ candidates yields an expected reward that is provably no worse than the reward obtained from the exploitation phase alone, and may in fact be better. Therefore, only using the candidates from the exploitation phase is a suboptimal strategy.

$\square$

## B ADDITIONAL RESULTS

### B.1 DETAILS OF REWARD-GUIDED BINARY SEARCH: ALGORITHM & MOTIVATION

**Algorithm Description.** Reward-guided binary search extends classic binary search by leveraging a reward model to guide the search process (see Algorithm 1 for pseudocode). At each step, instead of simply splitting the interval in half, the algorithm first queries the reward at $n$ candidate points

within the current interval. The reward is designed as the inverse of the distance to the target, possibly perturbed by noise to reflect real-world model uncertainty. Crucially, the algorithm selects the candidate point with the highest observed reward to refine the search interval, which similar to the strategy in test-time calibration, where high-reward completions are used as anchor points to guide subsequent exploration. This approach fundamentally alters the sampling distribution: rather than always bisecting the interval, the search adaptively concentrates queries near regions with higher estimated reward, dynamically steering the search direction. As a result, reward feedback not only accelerates convergence but also enables more efficient and adaptive exploration, especially when the reward model is reliable.

**Reward Model and Noise.** The reward function is defined as $r(x) = \frac{1}{|x-t|+1}$ where $t$ is the unknown target. In practice, the reward model may be noisy due to imperfect estimation, so we add Gaussian noise: $r_{\text{obs}}(x) = r(x) + \epsilon, \quad \epsilon \sim \mathcal{N}(0, \sigma^2)$. This noise models the fact that real-world reward models are not perfectly accurate and may deviate from the ground truth, making the search more challenging and realistic.

**Motivation.** Calibration (i.e., reward guiding) before each search step allows the algorithm to more efficiently narrow down the search space, especially when the reward model is reliable. Importantly, calibration fundamentally alters the sampling distribution: instead of always splitting the interval at the midpoint, the algorithm adaptively selects query points based on reward feedback, concentrating samples near regions with higher estimated reward. As shown in our experiments, increasing the number of calibration queries $n$ can dramatically reduce the number of search steps required, even under moderate noise. This demonstrates the practical value of reward-guided search for tasks like TTS, where reward feedback reshapes the sampling distribution and accelerates convergence.

---

**Algorithm 1:** Reward-Guided Binary Search with Calibration

---

**Input:** Search domain $[L, H]$, target $t$, calibration count $n$, reward noise $\sigma$
**Output:** Estimated target position
**while** $L < H$ **do**
    **if** $n > 0$ **then**
        Select $n$ evenly spaced probe points $\{x_1, \ldots, x_n\}$ in $[L, H]$;
        **foreach** $x_i$ **do**
            Query reward: $r_i = \frac{1}{|x_i - t| + 1} + \epsilon_i, \epsilon_i \sim \mathcal{N}(0, \sigma^2)$;
        Let $x^* = \arg\max_{x_i} r_i$;
        Estimate conservative bracket $[L', H']$ around $x^*$ using reward inversion and safety
         margin;
        $L \leftarrow \max(L, L')$;
        $H \leftarrow \min(H, H')$;
        Set comparison point $x_c = \lfloor (L + H)/2 \rfloor$;
    **else**
        Set comparison point $x_c = \lfloor (L + H)/2 \rfloor$;
    **if** $x_c < t$ **then**
        $L \leftarrow x_c + 1$;
    **else**
        $H \leftarrow x_c$;
**return** $L$

---

## B.2 FULL EXPERIMENTAL RESULTS

In this subsection, we provide the full experimental results for the MATH-500 benchmark, complementing the summary tables in the main text.

Table 6 reports the total runtime (seconds) for each model under the same setup, comparing uncalibrated Best-of-$N$ and CarBoN methods across different rollout budget $N$. Importantly, CarBoN achieves higher accuracy at smaller rollout budgets while maintaining competitive total runtime.

**Table 6: Total runtime (seconds) of four models on the MATH-500 benchmark, comparing Weighted Best-of-$N$ methods before and after calibration.** CarBoN achieves higher accuracy at lower rollout budgets, while maintaining comparable or faster total runtime relative to the corresponding uncalibrated Best-of-$N$: for example, $N = 64$ with CarBoN outperforms the uncalibrated Best-of-$N$ at $N = 256$ in accuracy.

| Model | Method | 8 | 16 | 32 | 64 | 128 | 256 |
|---|---|---|---|---|---|---|---|
| | | | | | N | | |
| Llama-3.2-1B-Instruct | Best-of-$N$ | 4.26 | 5.46 | 7.20 | 10.76 | 17.66 | 30.30 |
| | CarBoN | 6.27 | 11.09 | 16.82 | 34.47 | 47.98 | 82.15 |
| Llama-3.2-3B-Instruct | Best-of-$N$ | 5.06 | 7.70 | 9.68 | 14.08 | 19.14 | 32.48 |
| | CarBoN | 7.36 | 13.33 | 19.29 | 29.74 | 49.27 | 88.92 |
| Qwen2.5-1.5B-Instruct | Best-of-$N$ | 5.44 | 10.92 | 15.10 | 17.00 | 25.72 | 44.14 |
| | CarBoN | 8.25 | 14.00 | 20.59 | 31.54 | 55.23 | 99.59 |
| Qwen2.5-Math-1.5B-Instruct | Best-of-$N$ | 7.80 | 9.54 | 15.32 | 16.22 | 25.62 | 46.22 |
| | CarBoN | 9.39 | 12.49 | 20.93 | 27.19 | 60.56 | 113.05 |

For $N = 64$, CarBoN surpasses the uncalibrated Best-of-$N$ at $N = 256$ in accuracy for all models. Compared to the corresponding Best-of-$N$, the total runtime with CarBoN is lower for three of the four models, with the largest reduction for Qwen2.5-Math-1.5B-Instruct (27.19 sec vs. 46.22 sec), while for Llama-3.2-1B-Instruct the runtime is slightly higher (34.47 sec vs. 30.30 sec).

Table 7 reports the corresponding accuracy results for both Vanilla and Weighted decoding across all four models and different $N$. For small $N$, Vanilla selection can occasionally achieve the highest accuracy, but for larger $N$ (128, 256), CarBoN consistently outperforms other methods, showing stable gains under both Vanilla and Weighted selection.

Table 8 presents the ablation study for calibration parameters $(\delta, T)$ on two models (Llama-3.2-1B-Instruct and Qwen2.5-Math-1.5B-Instruct). Each experiment includes cases with only $\delta$, only $T$, or both combined (CarBoN), under both Vanilla and Weighted selection. While Vanilla selection occasionally achieves the highest accuracy for specific $N$ with a single parameter, Weighted decoding consistently performs best when combining $\delta$ and $T$, highlighting the complementary benefits of the two calibration parameters. All values are reported as accuracy (%).

**Table 7: Accuracy (%) of four models on the MATH-500 benchmark, comparing Vanilla and Weighted Best-of-$N$ methods before and after calibration.** CarBoN enables further improvements beyond the plateau of standard Best-of-$N$, with calibrated accuracy at $N = 64$ exceeding the uncalibrated results at $N = 256$, corresponding to up to $4\times$ less rollout budgets. Bold numbers indicate the better accuracy *within the same method type* (Vanilla vs. Vanilla, Weighted vs. Weighted) for each $N$.

| Model | Method | | 8 | 16 | 32 | 64 | 128 | 256 |
|---|---|---|---|---|---|---|---|---|
| | | | | | | N | | |
| Llama-3.2-1B-Instruct | Best-of-$N$ | Vanilla | **44.2** | 45.8 | 47.6 | **49.4** | 49.8 | 50.0 |
| | | Weighted | 42.0 | 44.6 | 47.8 | 48.6 | 50.6 | 50.8 |
| | CarBoN | Vanilla | 43.8 | **47.4** | **48.0** | 49.0 | **50.0** | **50.6** |
| | | Weighted | **43.0** | 45.6 | **48.4** | **51.0** | **51.8** | **51.8** |
| Llama-3.2-3B-Instruct | Best-of-$N$ | Vanilla | 56.8 | **59.0** | 60.0 | 60.2 | 60.4 | 60.6 |
| | | Weighted | 56.8 | 58.2 | 59.6 | 61.6 | 61.8 | 62.2 |
| | CarBoN | Vanilla | 57.0 | 58.8 | **61.0** | **61.6** | **61.6** | **61.8** |
| | | Weighted | **57.6** | **59.0** | **60.8** | **62.2** | **63.2** | **63.4** |
| Qwen2.5-1.5B-Instruct | Best-of-$N$ | Vanilla | **53.6** | 54.4 | 55.2 | 55.8 | 56.2 | 56.0 |
| | | Weighted | **56.4** | 57.6 | 61.4 | 62.0 | 62.6 | 62.2 |
| | CarBoN | Vanilla | 50.8 | 53.8 | 53.8 | **56.8** | **56.6** | **59.6** |
| | | Weighted | 55.0 | **60.0** | **61.8** | **62.4** | **64.0** | **64.4** |
| Qwen2.5-Math-1.5B-Instruct | Best-of-$N$ | Vanilla | **71.0** | 70.8 | 71.0 | 70.8 | 70.8 | 70.8 |
| | | Weighted | 73.6 | 75.4 | 76.4 | 75.6 | 76.4 | 76.8 |
| | CarBoN | Vanilla | 70.4 | **71.2** | **73.0** | **72.6** | **73.4** | **73.4** |
| | | Weighted | **74.2** | **76.0** | 76.4 | **77.2** | **77.2** | **77.8** |

**Table 8: Ablation study on calibration parameters $(\delta, T)$ and their combination (CarBoN) for Best-of-$N$ search on MATH-500.** We compare vanilla Best-of-$N$, applying a shift $(\delta)$, a temperature scaling $(T)$, and their joint calibration (CarBoN), under both vanilla and weighted selection strategies. All values report accuracy (%). Results show that CarBoN consistently improves accuracy across different $N$, especially with the weighted variant, highlighting the complementary benefits of $\delta$ and $T$. Bold numbers indicate the better accuracy *within the same method type* (Vanilla vs. Vanilla, Weighted vs. Weighted) for each $N$.

| Model | Method | | N | | | | | |
|---|---|---|---|---|---|---|---|---|
| | | | 8 | 16 | 32 | 64 | 128 | 256 |
| Llama-3.2-1B-Instruct | Best-of-$N$ | Vanilla | **44.2** | 45.8 | 47.6 | **49.4** | 49.8 | 50.0 |
| | | Weighted | 42.0 | 44.6 | 47.8 | 48.6 | 50.6 | 50.8 |
| | Best-of-$N$ w/ $\delta$ | Vanilla | 42.8 | 46.6 | **48.8** | 48.2 | 49.6 | 50.0 |
| | | Weighted | 41.8 | 43.8 | 48.2 | 49.0 | 51.0 | 51.2 |
| | Best-of-$N$ w/ $T$ | Vanilla | 42.4 | **47.6** | 46.8 | 48.4 | **50.4** | 50.4 |
| | | Weighted | 42.0 | 45.0 | 47.0 | 49.6 | 49.8 | 50.6 |
| | CarBoN | Vanilla | 43.8 | 47.4 | 48.0 | 49.0 | 50.0 | **50.6** |
| | | Weighted | **43.0** | **45.6** | **48.4** | **51.0** | **51.8** | **51.8** |
| Qwen2.5-Math-1.5B-Instruct | Best-of-$N$ | Vanilla | **71.0** | 70.8 | 71.0 | 70.8 | 70.8 | 70.8 |
| | | Weighted | 73.6 | 75.4 | **76.4** | 75.6 | 76.4 | 76.8 |
| | Best-of-$N$ w/ $\delta$ | Vanilla | **71.0** | 71.0 | 71.2 | 71.0 | 70.8 | 70.8 |
| | | Weighted | 74.2 | 74.8 | 75.6 | 77.0 | 76.6 | 77.0 |
| | Best-of-$N$ w/ $T$ | Vanilla | 70.0 | **71.2** | 71.0 | 70.8 | 70.8 | 70.8 |
| | | Weighted | 73.2 | 75.2 | 76.0 | 76.4 | 76.0 | 76.6 |
| | CarBoN | Vanilla | 70.4 | **71.2** | **73.0** | **72.6** | **73.4** | **73.4** |
| | | Weighted | **74.2** | **76.0** | **76.4** | **77.2** | **77.2** | **77.8** |

**Table 9: Pass@1 accuracy on larger closed-source and open-source models.** Evaluated under the same setup as in the main experiments (system prompt and $T = 0.8$), these results serve as reference points for comparing test-time scaling with smaller models.

| Type | Model | MATH-500 | AIME-2024 |
|---|---|---|---|
| Closed Source LLMs | gpt-5-nano | – | 11/30 |
| | o1-preview | 87.0% | 12/30 |
| | gpt-4o | 77.0% | 4/30 |
| Open Source LLMs | Qwen2.5-Math-7B-Instruct | 73.2% | 4/30 |
| | Qwen2.5-7B-Instruct | 61.4% | 3/30 |
| | LLaMA-3.1-8B-Instruct | 41.4% | 3/30 |

### B.3 SUPPLEMENTARY RESULTS ON LARGER MODELS (PASS@1)

For reference, we additionally report results on several larger closed-source and open-source models, evaluated under the same setup as in the main experiments (identical system prompt as shown in Appendix C.4 and sampling temperature $T = 0.8$). These results provide additional context, illustrating that test-time scaling with smaller models can approach the performance level of substantially larger models. Table 9 reports single-sample (@1) accuracies for the selected models.

### B.4 TEMPERATURE SCALING WITH SAMPLE SIZE $N$

We further investigate how the learned calibration temperature varies with the sample size $N$. Figure 4 plots the temperature across five difficulty levels as $N$ increases. Two consistent trends emerge: (i) more difficult problems require higher temperatures, as discussed in the main text, and (ii) larger $N$ also leads to higher optimal temperatures. The latter reflects that with more samples, a higher temperature is necessary to encourage sufficient diversity and thereby better utilize the expanded inference budget. Otherwise, generating many samples under a low temperature yields near-identical outputs, effectively wasting the additional budget. A simple example is buying 100 lottery tickets with the same number (low $T$), where even with more tickets the outcome remains largely unchanged. With diverse numbers (high $T$), additional tickets meaningfully increase the chance of winning.

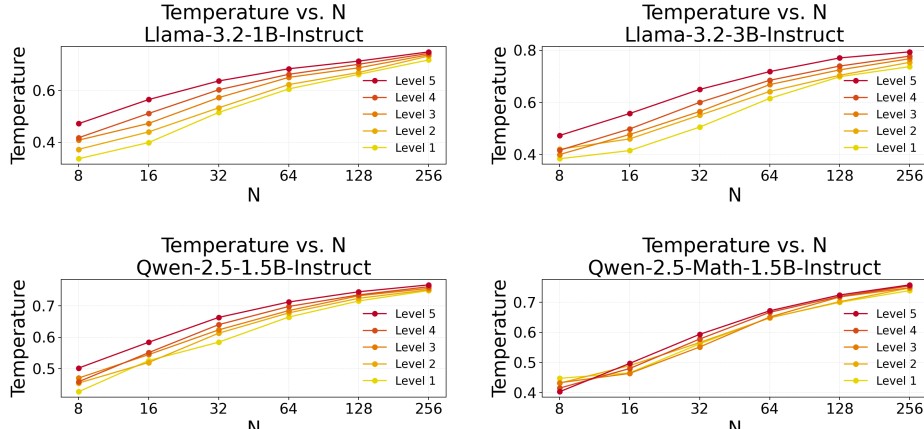

**Figure 4: Learned calibration temperatures across different rollout budget** $N$ **and problem difficulty levels.** Both larger $N$ and higher difficulty consistently lead to higher $T$, reflecting the increased diversity of top-$k$ completions. This adaptive scaling complements our earlier grid-search results in Appendix C.2, where small $N$ favored lower temperatures but larger $N$ required higher temperatures to fully leverage the broader exploration.

**Table 10:** Spearman rank correlations ($\rho$) in the best-of-$N$ setting (rollout budget $N = 256$), between problem difficulty and calibration dataset entropy from top-$k$ completions ($N_1 = 128$, $k = 32$ for calibration dataset), and learned calibration temperature ($N_2 = 128$).

| Model | Entropy | Temperature |
|---|---|---|
| Llama-3.2-1B-Instruct | 0.999 | 0.999 |
| Llama-3.2-3B-Instruct | 0.999 | 0.999 |
| Qwen2.5-1.5B-Instruct | 0.899 | 0.999 |
| Qwen2.5-Math-1.5B-Instruct | 0.999 | 0.999 |

This finding aligns with our earlier grid-search experiments (Appendix C.2). For small $N$, lower temperatures tend to reach peak accuracy earlier, while higher temperatures show little improvement initially but yield greater gains for larger $N$. This indicates that temperature affects the growth rate of accuracy, and that a fixed temperature may either converge early (low $T$) or show delayed benefits (high $T$), highlighting the need to adapt $T$ to $N$. Importantly, adapting $T$ to $N$ also explains why baseline methods with a fixed temperature tend to converge sooner than CarBoN, while CarBoN continues to improve as $N$ increases. Together, these results highlight that temperature is not a fixed hyperparameter, but should adapt naturally to both task difficulty and inference-time compute.

### B.5 CORRELATION BETWEEN PROBLEM DIFFICULTY AND CALIBRATION STATISTICS

To further examine how calibration relates to problem difficulty, we computed Spearman rank correlations ($\rho$) in our best-of-$N$ study with $N = 256$ setting, where $N_1 = 128$ samples are generated for exploration (select top-$k$ score completions to construct the calibration dataset), and $N_2 = 128$ samples are used in the second phase for exploitation.

Specifically, we considered two quantities: (i) the entropy of the calibration dataset constructed from top-$k$ completions in the exploration phase ($N_1$), and (ii) the learned temperature estimated from the calibration dataset and applied in the exploitation phase ($N_2$).

As shown in Table 10, the learned temperature exhibits an almost perfect correlation with problem difficulty across all four models ($\rho \approx 0.99999$), $p < 10^{-5}$). For the calibration dataset entropy, three models exhibit near-perfect correlations, while Qwen2.5-1.5B-Instruct shows a slightly weaker yet still strong correlation ($\rho \approx 0.9$, $p \approx 0.037$).

### B.6 Top-$k$ Token Overlap Metrics for High-Scoring Answers

We formalize four token-level overlap metrics that quantify how closely a group of generated answers (either after calibration with $\delta$ or an uncalibrated set) aligns lexically with the vocabulary used by the top-$k$ highest-scoring answers for each problem. All metrics are computed independently per problem and then macro-averaged across the full dataset (i.e., each problem contributes equally regardless of length).

Let Target denote the de-duplicated set of token IDs appearing in the union of the top-$k$ high-scoring answers for a given problem with special tokens removed. Let $X$ be the comparison token set built from either (i) calibrated generations after applying $delta$ ("calibration w/ $\delta$") or (ii) the uncalibrated ("No Calibration"). Define:

$$I = |\text{Target} \cap X|, \quad U = |\text{Target} \cup X|, \quad n_{\text{Target}} = |\text{Target}|, \quad n_X = |X|.$$

We report:

- Jaccard similarity: $J(\text{Target}, X) = \frac{|\text{Target} \cap X|}{|\text{Target} \cup X|} = \frac{I}{U}$.

- Dice (Sørensen–Dice) coefficient: $D(\text{Target}, X) = \frac{2|\text{Target} \cap X|}{|\text{Target}| + |X|} = \frac{2I}{n_{\text{Target}} + n_X} = \frac{2J}{1+J}$.

- Recall (coverage of high-scoring tokens): $\text{Recall}(\text{Target} \rightarrow X) = \frac{|\text{Target} \cap X|}{|\text{Target}|} = \frac{I}{n_{\text{Target}}}$.

- Precision (specificity toward high-scoring tokens): $\text{Precision}(\text{Target} \leftarrow X) = \frac{|\text{Target} \cap X|}{|X|} = \frac{I}{n_X}$.

**Interpretation.** Jaccard and Dice provide set-level similarity that penalizes both omissions (missing reference tokens) and additions (extra tokens outside the reference). Recall measures how completely the high-quality lexical signal is covered by the generated group, while Precision measures how selectively the group reuses only that high-quality signal (penalizing off-pattern or noisy additions). A calibration that increases Jaccard/Dice and Precision while maintaining high Recall indicates convergence toward the lexical core of high-scoring answers without excessive loss of useful diversity.

All four metrics are first computed per problem and then averaged uniformly across all problems (macro average). This prevents problems with larger token sets from dominating the aggregate.

## C Experiment Details

### C.1 Computational Environment

Experiments were conducted on two types of nodes: (i) two nodes with 4 × NVIDIA H100 (80GB) GPUs, 96 CPU cores, and 1 TB RAM; and (ii) two nodes with 8 × NVIDIA RTX 3090 (24GB) GPUs, up to 44 CPU cores, and 768 GB RAM. All experiments used Python 3.11.11, PyTorch 2.4.0, vLLM 0.6.3, and CUDA 12.9.

### C.2 Temperature Grid Search.

Figure 5 shows the results of Llama-3.2-1B-Instruct on the MATH-500 dataset using different temperatures ($T \in [0.1, 1.6]$ with step size 0.1)) for best-of-$N$ inference, with majority voting (left), naive (middle), and weighted (right) selection strategies. Across all settings of $N = 1, 2, 4, \ldots, 64$, lower temperatures (blue curves) consistently yield higher accuracy by focusing generation on high-quality answers. However, when the temperature is too low, the improvement with increasing $N$ becomes marginal, especially for majority voting and weighted selection, indicating that excessive concentration limits diversity and exploration. Conversely, higher temperatures (red curves) increase diversity but reduce accuracy for mathematical problems; at extremely high temperatures (e.g., $T = 1.6$), the model struggles to solve the tasks and accuracy drops sharply, highlighting the importance of careful temperature tuning. Previous findings (Beeching et al.) show that sampling with $T = 1.0$ sometimes leads the model to unexpectedly generate Chinese characters mid-solution

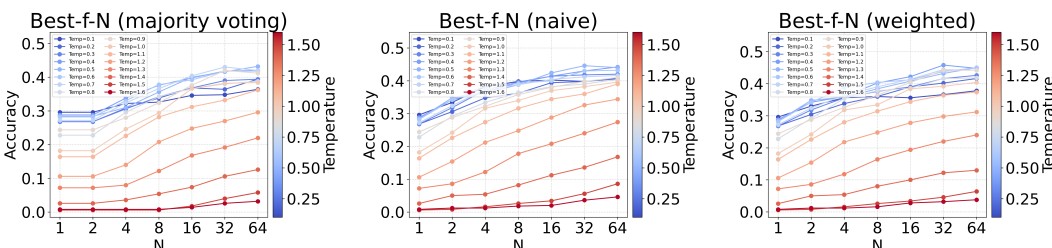

**Figure 5:** Results of Llama-3.2-1B-Instruct on MATH-500 with different temperatures $T$ for best-of-$N$ inference. From left to right: majority voting, naive, and weighted selection. Blue curves indicate lower temperatures, red curves indicate higher temperatures. Lower temperatures improve accuracy, but overly low temperatures limit diversity and the benefit of increasing $N$.

and hurts performance. Considering both accuracy and exploration, and following previous work (Snell et al., 2024), we adopt $T = 0.8$ as the baseline temperature for all experiments.

### C.3 CALIBRATION TRAINING DETAILS.

Calibration parameters $(\delta, T)$ are optimized using AdamW in a full-batch setting for 100 epochs per problem. The learning rate is $0.001$ with a constant schedule, and a weight decay of $10^{-2}$ is applied only to $\delta$. The parameters are initialized as $\delta = 0$ and $T = 0.8$. The loss is the negative log-likelihood over the top-$k$ candidates.

For each evaluation budget $N$, we split it evenly into $N_1 = N_2 = N/2$. In the first stage, $N_1$ completions are generated at $T = 0.8$, and the top-$k$ highest-scoring completions ($k = N_1/4$) form the calibration dataset. The calibration parameters $(\delta, T)$ are then trained directly on cached logits. In the second stage, the remaining $N_2$ completions are generated using the learned $(\delta, T)$.

This procedure ensures lightweight test-time calibration, requiring no additional model forward passes beyond the initial generation.

### C.4 SYSTEM PROMPT

For all test-time scaling experiments, we follow previous work Snell et al. (2024) and adopt the same system prompt across all models to ensure consistency and comparability of results. This allows us to isolate the effects of the scaling methods without introducing variability from different prompts, as detailed in Table 11.

**Table 11:** System Prompt for all experiments

---

**System Prompt**

---

Solve the following math problem efficiently and clearly:
- For simple problems (2 steps or fewer):
Provide a concise solution with minimal explanation.
- For complex problems (3 steps or more):
Use this step-by-step format:
## Step 1: [Concise description]
[Brief explanation and calculations]
## Step 2: [Concise description]
[Brief explanation and calculations]
...
Regardless of the approach, always conclude with:
Therefore, the final answer is: $\\boxed{answer}$. I hope it is correct.
Where [answer] is just the final number or expression that solves the problem.

---

