# OpenReview forum: "CarBoN: Calibrated Best-of-N Sampling Improves Test-time Reasoning"
_ICLR.cc/2026/Conference — ICLR 2026 Conference Withdrawn Submission_

### Official Review · Reviewer_Kako · 2025-10-31

**Soundness:** 1
**Presentation:** 2
**Contribution:** 1
**Rating:** 2
**Confidence:** 4

**Summary:**

CarBoN introduces test-time calibration for LLM reasoning, inspired by the observation that Best-of-N sampling discards N-1 trajectories after selecting the highest-scoring output. The method splits inference budget N into exploration and exploitation phases, reusing high-scoring discarded samples to learn input-specific calibration parameters (temperature T and additive shift δ) that guide subsequent generation toward more reliable reasoning paths.

**Strengths:**

- Clever reuse of discarded trajectories
- Ablations isolate T and δ contributions

**Weaknesses:**

- Theorem 2 assumes unique optimal output y*, unrealistic for mathematical reasoning with multiple valid solution paths.
- The "4× fewer rollouts" compares CarBoN@N=64 accuracy vs baseline@N=256 accuracy—conflating sample efficiency with computational efficiency. Both methods use N PRM calls for their respective budgets.
- Severely outdated experimental setup: Models are almost a year old + MATH500 is a weak evaluation dataset at this point. Even if this paper gets in, by ICLR 2026, these models would be almost 18+ months old.
- The calibration depends on the PRM quality, and PRMs are themselves unreliable, especially for reasoning models.
- Single PRM across all experiments. We don't know how the quality of PRM can affect the empirical results.


Minor nit:
- Figure 5: Title has "of" spelled as "f"

**Questions:**

- Does this work on current reasoning models, such as Qwen3?
- The dependence on PRMs is not ideal. Any ideas on how to get rid of this dependence?

---

### Official Review · Reviewer_RJpm · 2025-11-01

**Soundness:** 2
**Presentation:** 2
**Contribution:** 2
**Rating:** 2
**Confidence:** 3

**Summary:**

This paper introduces a new method to improve test-time reasoning through a two-phase process. In the first phase, it performs naïve parallel sampling with a fixed temperature. Then, it trains two lightweight parameters: one is the temperature, and the other is a bias vector before the language model head, to recalibrate the output distribution for each problem. After that, it runs another round of parallel sampling using this updated distribution, and finally selects the answer. This method relies on a continuous reward model to rank the responses. Additionally, the paper provides a theoretical argument for why this recalibration works and presents empirical evidence showing that the method improves upon the naive parallel sampling baseline by one or two points across benchmarks.

**Strengths:**

1. The proposed method is straightforward, making it easy to implement and test.

2. The paper provides detailed empirical results across multiple models from various open-source families.

3. The paper discusses how to combine the proposed method with more complex token-level sampling strategies beyond just naïve parallel sampling.

4. The paper offers an interesting analysis showing that the learned temperature is higher for more difficult problems, which aligns well with the intuition behind the design.

**Weaknesses:**

1.  The theory statement provided in the paper is not very related to the empirical result. The major result Theorem 1 essentially state that if the probability of the unique correct solution increases after training, then Pass@N will increase as well. This statement is rather trivial and is true for any method essentially.

2. The paper tested the methods on some small datasets such as AIME that only contains 30 problems. However, the paper does not provide the average performance over multiple random seeds. This makes it hard to judge whether the improvement of 1 problem is statistically significant. For some larger datasets like MATH, the paper didn't provide error bars.

3. The temperature chosen for baseline is tuned on a LLaMA 1B model. It is not clear that whether this temperature will be suboptimal for Qwen models tested.

4. [Minor] The synthetic example in Figure 1 is not very easy to understand as the configuration is not described in detail in the main paper.

**Questions:**

1. See weakness 2, can the authors provide some qualitative estimation of the randomness to better justify the significance of the experiments?

2. What is the typical wall time for training $T$ and $\delta$ for every problem. How does this compare with the sampling time?

---

### Official Review · Reviewer_soou · 2025-11-04

**Soundness:** 2
**Presentation:** 3
**Contribution:** 3
**Rating:** 2
**Confidence:** 4

**Summary:**

The paper introduces calibrated sampling for language model to push the model's output distribution towards more promising outputs. The method modifies the logits via adding a linearly transformed vector and choosing a new sampling temperature, which are optimized through a simple optimization. The paper applies this technique to Best-of-N by increasing the chance of generating the top results from the exploration phase, and observes accuracy improvements. Some theoretical justifications are provided.

**Strengths:**

1. I find the idea and the technique fascinating and believe it might have good potential. The calibration technique is a simple way to shift the output distribution of the LLM towards more desirable outputs, without the need for any training.

2. The paper is generally easy to follow, though I find some theoretical parts too vague.

3. I like the analysis of calibrated temperature for various difficulty levels.

**Weaknesses:**

1. My main concern about the paper is the limited breadth and rigor of the experiments:

1A: The AIME results are not informative at all as they seem to be a single run on only 30 questions. It is essential to perform multiple independent runs and report the average accuracy percentage to have a meaningful figure. The current results are too noisy to make any conclusions from.

1B: With unreliable AIME results, the paper's evidence comes down to only one benchmark. I find it necessary to test the method on a variety of benchmarks, ideally on different subjects and final answer styles (multiple choice, short answer, open ended, etc). I am concerned some special property MATH questions might have been helpful for these results. The additive vector $\delta$ is effectively a shift in the final hidden space and it might behave differently. It is perhaps ok, but it needs to be tested to have a conclusive test of effectiveness.

1C: It is also important to report error bars to understand the evaluation is accurate enough for the drawn conclusion. I expect for MATH500, this is just a matter of reporting, but perhaps AIME will need more runs.

2. Unfortunately, the theoretical results provide little to no support for the algorithm's promise. I understand that the main contribution of the paper is empirical. However, the theoretical results carry very little information and were mostly a distraction from the paper's strengths.

2A: The theorem statement is very weak. The statement is on a very loose lower bound and under strong explicit (mentioned in the thorem) and implicit assumptions. The implicit one being the fact that the goal is to sample the highest reward, which is only true under perfect rewards. Moreover, most of the theorem result is based on "calibration increases the probability of $y^*$".  This is a very strong assumption.

2B: The statement of Lemma 1 (both in main text and appendix) uses new complicated quantities without their formal definition and is hard to parse. Nonetheless, the result does not seem to be surprising. The fixed parameters are expectedly not the ones that create the highest chance of generating $D_{calib}$. I like the insight as a motivation for the algorithm, though.

3: As another path to improve the paper, I would suggest investigate the underlying dynamics more. The vector $\delta$ is a shift in hidden space and some more visualizations may be possible. Currently, only temperature is discussed.

4: Further discussions on the choice of hyperparameters is welcome: $\lambda_\delta$, $k$, $N_1$, $N_2$

**Questions:**

Please see weaknesses. One additional questions:

What do you think about using the rewards in the exploration phase as the weights of the outputs in calibration loss, instead of using them just for choosing which outputs to use.

---

### Note · Authors · 2025-12-02

**Comment:**

Dear Reviewers,

We appreciate the effort you have dedicated to reviewing our submission. After careful consideration, we have decided to withdraw the paper to clarify the contribution.

We would like to take this opportunity to briefly clarify some technical points:

- Theory: Lemma 1 shows that test-time calibration can increase the probability of sampling high-reward outputs. Theorem 2 then establishes that this increase leads to a strictly higher expected Best-of-N reward lower bound. In practice, the ‘best output’ is represented through the PRM.

- Role of the PRM: The PRM is used solely to rank candidate outputs. Calibration guides the model toward generating higher-quality outputs based on the selected high-ranking samples, rather than directly using reward scores.

- Sampling efficiency: The comparison of CarBoN@N=64 vs baseline@N=256 demonstrates that the same or higher accuracy can be achieved with fewer sampled trajectories, highlighting sample efficiency.

We hope these clarifications provide useful context, and we thank the reviewers again for their time and  feedback.

**Withdrawal Confirmation:**

I have read and agree with the venue's withdrawal policy on behalf of myself and my co-authors.